# The Use of Electrochemical Voltammetric Techniques and High-Pressure Liquid Chromatography to Evaluate Conjugation Efficiency of Multiple Sclerosis Peptide-Carrier Conjugates

**DOI:** 10.3390/brainsci10090577

**Published:** 2020-08-21

**Authors:** Efstathios Deskoulidis, Sousana Petrouli, Vasso Apostolopoulos, John Matsoukas, Emmanuel Topoglidis

**Affiliations:** 1Materials Science Department, University of Patras, 26504 Patras, Greece; stathis.deskou@gmail.com (E.D.); sousanapetr@gmail.com (S.P.); 2Institute for Health and Sport, Victoria University, Melbourne, VIC 3030, Australia; vasso.apostolopoulos@mail.com; 3Newdrug, Patras Science Park, 26500 Patras, Greece; 4Department of Physiology and Pharmacology, Cumming School of Medicine, University of Calgary, Alberta, AB T2N 4N1, Canada

**Keywords:** mannan, peptide, conjugation, MOG_35-55_, Graphite/SiO_2_ electrode, voltammetry, HPLC, multiple sclerosis, immunotherapy, vaccine

## Abstract

Recent studies have shown the ability of electrochemical methods to sense and determine, even at very low concentrations, the presence and quantity of molecules or analytes including pharmaceutical samples. Furthermore, analytical methods, such as high-pressure liquid chromatography (HPLC), can also detect the presence and quantity of peptides at very low concentrations, in a simple, fast, and efficient way, which allows the monitoring of conjugation reactions and its completion. Graphite/SiO_2_ film electrodes and HPLC methods were previously shown by our group to be efficient to detect drug molecules, such as losartan. We now use these methods to detect the conjugation efficiency of a peptide from the immunogenic region of myelin oligodendrocyte to a carrier, mannan. The HPLC method furthermore confirms the stability of the peptide with time in a simple one pot procedure. Our study provides a general method to monitor, sense and detect the presence of peptides by effectively confirming the conjugation efficiency. Such methods can be used when designing conjugates as potential immunotherapeutics in the treatment of diseases, including multiple sclerosis.

## 1. Introduction

Voltammetric techniques, including differential pulse voltammetry (DPV) and cyclic voltammetry (CV), as well as high-performance liquid chromatography (HPLC), were applied to identify and detect a peptide to its conjugated carrier. This study describes for the first time an alternative, fast, low cost and reliable method for the adequate and reliable determination of an active pharmaceutical ingredient (API) in the biocompatible matrix. The performance of the voltammetric techniques is strongly dependent on the performance of the working electrode used. Film electrodes, such as the graphite/SiO_2_ used in this study, are being used in electrochemistry, as it has a number of advantages over the standard metallic and glass carbon electrodes. These include ease of manufacture requiring lower temperatures, low cost, the high surface area that could be rapidly renovated, simple handling, and their increased conductivity in a wide range of potentials. In addition, these techniques exhibit a wide range of anodic and cathodic peaks and great electrocatalytic activity and stability. All these features are crucial for the correct choice of a working electrode, especially when direct electrochemistry is conducted [1]. We recently demonstrated that these film electrodes modified or not, could be used for electrochemical drug sensing, for validation in food chemistry, and for the immobilization of heme proteins for studying protein/electrode interaction [2,3,4]. The electrochemical analytical methods were recently applied effectively in the detection of anti-hypertensive drug losartan [3] and have applied this method to detect peptides in peptide-carrier conjugates. The peptide used was the multiple sclerosis (MS) immunogenic peptide from myelin oligodendrocyte (MOG_35-55_).

Numerous methods have been established for the analytical determination of drugs at low concentrations, using state of the art systems, such as HPLC, high-performance thin-layer chromatography and capillary electrophoresis/capillary electrochromatography [3]. Although these methods provide very accurate and reliable data, they are costly, time consuming, and involve the use of expensive equipment and consumables. In addition, sample pre-treatment is usually necessary. In this sense, electrochemical methods have emerged as low cost, reliable alternatives for the characterization of peptides and drugs. Different electrochemical techniques, involving voltammetry or potentiometry, have been implemented for drug analysis, as they offer ease of preparation and operation, high sensitivity, fast response time, high quantification and detection limits, reasonable selectivity, wide linear range, and are cost effective [3]. In this regard, we applied voltammetry techniques to monitor the conjugation of a peptide to its carrier, for the first time as a proof of concept study.

MS is regarded an autoimmune disease where immune cells (such as, Th1, Th17, macrophages, B cells) and their constituents (pro-inflammatory cytokines) are involved in the pathophysiology of the disease, with destruction of myelin sheath and loss of neurological function [5,6,7,8,9,10]. In an attempt to develop immunotherapeutics against MS using immunogenic/agonist peptides is to either alter the peptide to make it an antagonist [11,12,13,14,15,16], make it cyclic [17,18,19], or conjugate it to an appropriate carrier, which would deliver the peptide in such a manner to either induce tolerance, or alter the profile of T cells from pro-inflammatory (Th1) to anti-inflammatory (Th2) [13,14,15]. One approach which our team has developed, is to use mannan, a poly-mannose carrier conjugated to MS peptides [20,21,22,23,24]. This approach was developed over 25 years ago by the group of Apostolopoulos et al., to be effective in targeting peptides and proteins to dendritic cells in a number of different cancer vaccine models, some of which were translated to human clinical trials [25,26,27,28,29,30,31,32]. As such, mannan was used as a carrier and conjugated to immunodominant MS peptides including MBP_83-99_, PLP_139-141_, and MOG_35-55_ or their analogues, and were shown in animal models to tolerize T cells or switch Th1 cells to Th2 cells, depending on the peptide analogue used and showed stimulation of Th2 cells in peripheral blood mononuclear cells from patients with MS [17,21,33,34,35]. The conjugation of mannan, in its oxidized form (OM), to MOG_35-55_ peptide (MOG_35-55_ was used as an example in this study) via a (Lys-Gly)_5_ linker [(KG)_5_] was used and evaluated (OM-(KG)_5_-MOG_35-55_ conjugate) using voltammetric techniques [1,2,3,4,36,37,38]. The conjugation between OM and peptide (MOG_35-55_) occurs via formation of Schiff bases between the free amines of the linker (KG)_5_ and aldehydes of OM. The synthesis and efficacy of these conjugates have been described in numerous studies [23,24,39,40]. However, the extent of conjugation and the redox condition of the participating sugars, such as mannan, are most times assessed by high cost, complicated and lengthy analytical methods, such as capillary electrophoresis and polyacrylamide gel electrophoresis [23,24,39,40].

Among the approaches used in recent years for the immunomodulation of MS, the conjugation of mannan with myelin peptides has shown much promise, including that of OM-(KG)_5_-MOG_35-55_, which induces tolerance in mice, providing a promising conjugate for further studies. The electrochemical and HPLC analysis for identification of peptides or their mutants in mannan based conjugates requires specialized techniques, which differ significantly from those methods used for small molecules. In this study, novel analytical methods were developed and applied, that clearly, sense, detect, and confirm the conjugation of OM with MOG_35-55_. Further, this study makes it possible to accurately evaluate the stability of the peptide component in the conjugate using HPLC [41,42].

## 2. Materials and Methods

### 2.1. Materials

Sodium metasilicate (Na_2_SiO_3_) (SiO_2_, 50–53%), NaH_2_PO_4_, mannan isolated from yeast cells (*Saccharomyces cerevisiae*), potassium ferricyanide, ferrocyanide, and potassium chloride were obtained from Sigma Aldrich Chemie GmbH (Taufkirchen, Germany). MOG_35-55_ and MOG_37-55_ peptides were supplied by NewDrug S.A., Patras Science Park, Greece and purchased from China peptides Inc. The peptide analogue (Lys-Gly)_5_-MOG_35-55_, referred as (KG)_5_-MOG_35-55_, was synthesized using standard peptide chemistry techniques and previously published by our group. Briefly, Fmoc/tBu methodology was used which included 2-chlorotrityl chloride resin (CLTR-Cl) and N^a^-Fmoc (9-fluorenylmethyloxycarboxyl) side chain protected amino acids [43,44]. The purity of the peptides were shown to be >97% by analytical HPLC. Graphite powder (synthetic, APS 7–11 μm, 99%) was obtained from Alfa Aesar. Soda lime glass slides (75 mm × 25 mm × 1.1 mm), with 15 Ohm/sqr Indium Tin Oxide (ITO) coating were obtained from PsiOTec, UK. All chemicals were of analytical grade and used without the need for further purification. All solutions were prepared in deionized water with resistance R = 18 MΩ cm.

### 2.2. Graphite/SiO_2_ Film Electrodes Preparation

The graphite/SiO_2_ film electrodes were prepared as described [2,3]. Briefly, silicate liquid polymer (50% Na_2_SiO_3_; pH 12–13) was gently mixed with 20% graphite powder at 23 °C, until the mixture became homogeneous and acquired a “sticky” texture. The mixture underwent ultrasonication for 2 min for the graphite powder to be fully soluble, and 100 μL of the silicate/graphite suspension were applied on the surface of a conductive ITO glass slide using the “Doctor Blade” technique. Prior to the deposition of the silicate/graphite suspension, the ITO glass slides were cleaned in a detergent solution using an ultrasonic bath for 15 min, and then rinsed with 18 MΩ distilled water and ethanol. Each glass slide was masked with 3M Magic Scotch tape (thickness 62.5 μm; type 810), in order to control the width and the thickness of the mixture spread area. For each graphite/SiO_2_ film deposition, one layer of tape was used which provided a size 1 × 1 cm^2^ and film thickness of ~66 μm. The films were allowed to dry for 30 min in a class 4000 room, prior to placing them in a preheated oven (330 °C) for 100 min. If required, the liquid suspension could be stored in an insulated flask at 25 °C for later usage. The resulting ITO substrates with the deposited graphite/SiO_2_ films were cut in 10 mm × 25 mm pieces before use.

### 2.3. Characterization of Graphite/SiO_2_ Film Electrodes

Field emission scanning electron microscopy (FE-SEM) using an FEI inspect microscope (25 kV) was used to determine morphology and thickness of the Graphite/SiO_2_ film. The films were prepared by AU sputtering to increase the conductivity of the samples. Energy dispersive spectroscopy EDS was also used for the elemental analysis of the Graphite/SiO_2_/ITO films.

### 2.4. Preparation of (KG)_5_-MOG_35-55_ Peptide

MOG_35-55_ agonist peptide was synthesized in our labs, >97% purity, with (KG)_5_ extended at the N-terminus of the peptide. Peptide was prepared using our methods, either by coupling, catalyzed by microwave radiation in a CEM Liberty microwave system or by using the conventional step by step procedure by solid phase peptide methods (as described in [45]). (KG)_5_-MOG_35-55_ peptide was also purchased by China Peptides Inc. In house synthesized peptides and purchased peptides were confirmed by HPLC and Mass Spectroscopy for purity and identity.

### 2.5. Preparation of Oxidized Mannan

Mannan (14 mg) was dissolved in 1 mL phosphate buffer (0.1 M sodium phosphate, pH 6.0), and was oxidized using 0.1 M sodium periodate and incubated at 4 °C for 1 h, after which 10 μL ethanediol was added for 30 min at 4 °C. Oxidized mannan (OM) was passed through a PD-10 column (Sigma Aldrich Chemie) pre-equilibrated in sodium bicarbonate buffer (sodium carbonate: Sodium bicarbonate, pH 9.0). Two ml of OM fraction (7 mg/mL) was collected and kept in the dark.

### 2.6. Conjugation of Oxidized Mannan to Peptide

To the OM fraction (2 mL; 7 mg/mL, sodium bicarbonate pH 9.0 buffer), 1 mg of (KG)_5_-MOG_35-55_ peptide was added and allowed to react overnight in the dark at 23 °C. A list of peptides and conjugates are summarized in Table 1.

### 2.7. Monitoring of Conjugation by HPLC

We used a Waters 2695 HPLC (Alliance) system with a photodiode array detector equipped with a Lichrosorb RP-18 reversed phase analytical column (C18 35 μm, 4.6 × 50 mm PIN 186003034). Analysis was achieved with stepped linear gradient of solvent A (0.08% TFA in H_2_O) and in solvent B (0.08% TFA in 100% acetonitrile) for 30 min with a flow rate 3 mL/min. The conjugation of OM with (KG)_5_-MOG_35-55_ peptide was evaluated by HPLC. The (KG)_5_-MOG_35-55_ HPLC peak disappeared within six hours indicating completion of conjugation to OM.

### 2.8. Electrochemical/Electrocatalytic Measurements

Electrochemical measurements were conducted using an Autolab PGStat-101 potentiostat (Metrohm, Utrecht, The Netherlands). The electrochemical cell comprised of a 10 mL, three-electrode stirring glass cell with a Teflon cap, a platinum mesh flag as the counter electrode, a Ag/AgCl/KCl_sat_ reference electrode and a Graphite/SiO_2_ film on ITO conducting glass as the working electrode. The electrolyte contained a solution of NaH_2_PO_4_ (10 mM; pH 7.0), which was deoxygenated with argon prior to any measurements and an argon atmosphere was kept throughout the measurements. The DPV measurements took place in a potential range between −1 to +0.05 V. The optimized parameters of DPV correspond to a step potential at 5 mV, amplitude of 50 mV, modulation time of 25 ms with scan rate 100 mV s^−1^ and a frequency of 50 Hz. All potentials are reported against Ag/AgCl and all experiments were carried out at 23 °C.

## 3. Results and Discussion

### 3.1. FE-SEM Characterization

The general thickness and surface morphology of the graphite/SiO_2_ films were demonstrated by FE-SEM. The top-view of the FE-SEM image (Figure 1a) shows that the surface of the graphite/SiO_2_ film is rough and non-uniform with many wrinkles. It exhibits increased porosity and a high effective surface area. Figure 1b presents the cross section of a graphite/SiO_2_ film electrode, with an estimated film thickness of ~65 μm as set by the adhesive tape used; the EDS for a graphite/SiO_2_ film carried out during the FE-SEM analysis is shown in the Appendix A. The characteristic peaks of Na, O, and Si, due to the use of silicate glue (Na_2_SiO_3_), are presented in high intensity, thus, the peak of C is presented in lower intensity. Hence, the results validate the reduced concentration of carbon in the mixture used for the fabrication of the graphite/SiO_2_ films.

### 3.2. UV Characterazation of (KG)_5_-MOG_35-55_ Peptide with Increasing Amounts of OM

It is known that most peptides exhibit strong absorbance at around 280 nm, due to aromatic amino acids (tyrosine and tryptophan) or disulfide bonds in the peptide sequences [46,47]. Figure 2 shows the UV-vis spectra of (KG)_5_-MOG_35-55_ with increasing amounts of OM. The increase of absorbance at 280 nm confirms the conjugation of MOG_35-55_ peptide to OM. The intensity of the absorption peak at 280 increases until all of the free peptide in solution is conjugated to the OM. It should be noted that the conjugate of (KG)_5-_MOG_35-55_ with OM took place in solution and not on the surface of the graphite/SiO_2_ film electrode as due to its non-transparency it is impossible to monitor the conjugation process on its surface. All the UV-visible absorption spectra of the peptide was recorded using a Shimadzu UV-1800 spectrophotometer.

### 3.3. Electrochemical Analysis Showing Conjugation of (KG)_5_-MOG_35-55_ to OM

Electrochemical characteristics of the graphite/SiO_2_ film electrode were investigated by CV. Figure 3a shows the electrochemical behavior of a bare graphite/SiO_2_ film electrode in a solution of 0.1 M KCl and 5 mM of [Fe(CN)_6_]^3−/4−^ through CV in the potential range of +1 to −1 V at different scan rates. Figure 3b shows the currents (anodic and cathodic) from the plots of I vs. square root of scan rate (*v*^1/2^). Straight lines form for both the anodic and cathodic currents, confirming that a diffusional process has occurred in the reaction of ferrocyanide/ferricyanide. In addition, these results confirm that fast electron transfer occurs on the Graphite/SiO_2_ film electrode due to its increased conductivity and surface area. In order to calculate the electroactive surface area of the film electrode, the Randles-Sevcik equation was used [36]:(1)$$i_{p}=\left(2.69\times 10^{5}\right)\times A\times D^{1/2}\times n^{3/2}\;\times C\;\times \;v^{1/2}$$
where *i_p_* corresponds to the maximum current (in Amperes), *n* is the number of electrons transferred (*n* = 1), *D* is the diffusion coefficient (cm^2^ s^−1^) of [Fe(CN)_6_]^3−/4−^ solution (7.6 × 10^−6^ cm^2^ s^−1^) [37], *A* is the electrode area (cm^2^), *C* is the concentration (molcm^−3^) and *v* is the scan rate (mV s^−1^) and thus the electroactive surface area of the graphite/SiO_2_ was estimated to be 0.0039 cm^2^.

The electrochemical behavior of the graphite/SiO_2_ film electrode was then investigated in the presence and absence of the MS myelin epitope peptide vaccine (OM-(KG)5-MOG_35-55_). Figure 3c shows the effect of scan rate of a bare graphite/SiO_2_ electrode, before the detection of the OM-(KG)5-MOG_35-55_, at a scan rate range of 0.01 to 0.1 V s^−1^. All electrochemical experiments were performed in a peptide free, anaerobic 10 mM NaH_2_PO_4_ (pH 7.0). The bare graphite/SiO_2_ film electrode shows the characteristic charging/de-charging currents, and no cathodic or anodic peaks are observed even at the slowest scan rate (0.01 V s^−1^). One of the advantages of using graphite paste electrodes is the increased conductivity, which allows a broader study of redox reactions occurring at very high or low biases (ranging from +1 V to −1 V). Further, the slower scan rate applied, the smaller the resulting current is obtained. Figure 3b, on the other hand, showing the CVs of OM-(KG)_5_-MOG_35-55_ on the graphite/SiO_2_ film electrode, exhibits not only the characteristic charging/discharging currents assigned to electron injection into sub-band gap/conduction band states of the graphite/SiO_2_ electrode, but also two reduction peaks around −0.22 V and −0.67 V and a broad re-oxidation peak at −0.1 V.

The redox peak currents were shown to be proportional to the scan rate, characteristic of quasi-reversible behavior. The rate of reaction between the graphite/SiO_2_ electrode and the conjugate, OM-(KG)_5_-MOG_35-55_ was not fast enough to maintain equal concentrations of oxidized and reduced species at the surface of the electrode. In addition, the CV responses were shown to be stable, with the waveforms being unperturbed after being scanned several times, whilst no other consumption of the complex occurred nor other undesirable reactions in the phosphate buffer took place.

In Figure 3d, the two cathodic peaks at −0.27 V and −0.7 V and the wide anodic peak approximately at −0.1 V observed are due to the presence of the OM-(KG)_5_-MOG_35-55_. The two cathodic peaks correspond to the linker molecule (KG)_5_ used to conjugate the MOG_35-55_ peptide to OM, that contains 5 lysines and 5 glycines to its structure. Thus, the cathodic peaks attributed to the presence of lysines. On the other hand, the wide oxidation peak occurred probably due to superfluity of the free (KG)_5_-MOG_35-55_ peptide that was not able to conjugate to OM and created the final complex of the OM-(KG)_5_-MOG_35-55_ conjugate.

The CVs of the constituents of the OM-(KG)_5_-MOG_35-55_ conjugate are shown in Figure 4. According to Figure 4a, as mentioned earlier, the bare graphite/SiO_2_ film electrode exhibited no reduction or oxidation peaks which is consistent with the currents being limited by the graphite conductivity at the voltage biases reported herein. On the other hand, the CV of the film electrode in the presence of mannan in 0.1 M buffer exhibited an oxidation peak at approximately 0.5 V, and the CV of the film electrode in the presence of 0.002 mg/mL OM displayed a slight cathodic peak at −0.56 V and the characteristic anodic peak at −0.1 V. At the same time, the electrochemical behavior of peptides MOG_35-55_ and MOG_37-55_ were examined. The main difference between these two peptides is that the MOG_35-55_ peptide contained and additional linker with 5 lysines (KG)_5_, whilst the MOG_37-55_ peptide included a linker, which only contained 1 lysine. This was confirmed in Figure 4b, which displays the CVs of the Graphite/SiO_2_ film electrode in the presence of each peptide. The two cathodic and anodic peaks observed are due to the presence of the lysine residues, however, the CV scan of the MOG_35-55_ peptide exhibits a higher current and a wider electrochemical window compared to the CV scan of MOG_37-55_ peptide, as the latter contained only 1 lysine residue.

DPV is a more sensitive approach compared to CV and hence, has been extensively used as a more sensitive method for the detection of molecules in low concentration [38]. In Figure 5, the DPVs are recorded for the bare film electrode, as well as for each part that constitutes the final structure of OM-(KG)_5_-MOG_35-55_ conjugate on the Graphite/SiO_2_ working electrode. As can be seen in Figure 5a, the bare graphite/SiO_2_ is free of any redox peaks. However, in Figure 5b, there are two peaks which correspond to (KG)_5_-MOG_35-55_ peptide, approximately at −0.65 V and −0.27 V, respectively. Figure 5c shows the DPV of mannan (in 0.1 M phosphate buffer) on the surface of the film electrode, displaying a clear sharper peak at around −0.26 V. The last step in order to evaluate the conjugation of peptide (KG)_5_-MOG_35-55_ with OM via DPV measurements is depicted in Figure 5d with a clear and distinct peak at −0.28 V and a shoulder peak at −0.62 V, which are actually due to the presence of OM-(KG)_5_-MOG_35-55_ on the graphite/SiO_2_ film electrode (after the addition of 0.002 mg/mL of OM). This is a proof of concept study, and we intend to further study the quantification of this and other conjugates, focusing on the limit od detection (LOD) of these conjugates using voltammetric techniques.

### 3.4. Complete Conjugation between (KG)_5_-MOG_35-55_ Peptide to OM is Monitored by HPLC

Contrarily to the conjugation of MOG_35-55_ peptide with mannan, which did not occur, the reaction of (KG)_5_-MOG_35-55_ with mannan (oxidized or not) resulted in gradual conjugation of (KG)_5_-MOG_35-55_ peptide within 6 h depicted in the gradual loss of the HPLC peak during this period (Figure 6). The amino groups of lysine residues within (KG)_5_ forms a Schiff base reaction with the aldehyde groups of OM (resulting after the oxidation of mannan). The (KG)_5_-MOG_35-55_ peptide peak at 9.62 gradually disappears within this period, showing complete conjugation of (KG)_5_-MOG_35-55_ peptide to OM. Figure 6b shows the completion of conjugation within six hours.

### 3.5. The Importance of the Linker (KG)_5_ for Conjugation of Peptides to OM

The conjugation of MOG_35-55_ peptide to OM was achieved through (KG)_5_ linker, as previously described [23]. As demonstrated, this approach provides simple and efficient conjugation by the Schiff base reaction, where aldehyde groups of OM reacts with the amino groups of the lysine side chains of the (KG)_5_-MOG_35-55_, peptide. In previous similar studies using the linker KG of varying lengths, (KG)_n=1-5_, we noted that the length of the linker plays a crucial role in the ability of peptides to be efficiently conjugated to the OM scaffold [48].

### 3.6. Mannan-Peptide Conjugate

In the OM-(KG)_5_-MOG_35-55_ conjugate, unreacted aldehyde groups are necessary to immunoregulate the peptide to dendritic cells. This is a result of ethylene glycol addition to blockade further oxidation, and in line with previous studies on MUC1-mannan conjugates in cancer research, which required aldehyde groups in order to activate dendritic cells [39]. The matrix also contains intact mannose units, not oxidized, necessary to bind to the mannose receptor of the dendritic cells and their activation via toll-like receptor 4 [49,50,51,52]. In particular, the procedure we followed to produce the mannan-peptide conjugate allows: (i) the presence of antigen peptide MOG_35-55_ connected with aldehyde groups of the OM through immune bonds (Schiff base) with the amino groups of the lysine side chain in the (KG)_5_-MOG_35-55_ peptide. The peptide-OM conjugate is delivered to dendritic cells via the mannan scaffold for regulation of the immune system; (ii) the presence of unreacted aldehyde groups are necessary to modulate dendritic cells; and (iii) the presence intact mannose units, not oxidized, necessary to bind to the mannose receptor of the dendritic cells.

### 3.7. Chemistry of the Mannose Cleavage

The cis-diols can form a cyclic complex upon oxidation with strong oxidizing agents as periodate. This allows the cleavage of the bond between the two carbons bearing the two hydroxyl groups, leading to the formation of aldehyde groups. Mannose is a carbohydrate, which holds two hydroxyl groups at positions 2,3 of the ring in a cis- position. This allows the oxidizing agent sodium periodate to form a cyclic complex, which finally leads to cleavage of the carbon-carbon bond bearing the cis-hydroxyl groups. This complex cannot be formed if the hydroxyl groups at the adjacent carbon atoms are in a trans position and subsequently this carbon-carbon bond cannot be cleaved. The formation of the cyclic mannose-periodate complex is leading finally to the cleavage of the ring and the formation of the two aldehyde groups. These groups react with the amino groups of the five lysines of the (KG)_5_-MOG_35-55_ to form double bond imines (Schiff base reaction) thus, the MOG_35-55_ peptide attached to the mannan scaffold. Figure 7 shows the mechanism of cis diol cleavage.

## 4. Conclusions

We developed and confirm an analytical electrochemical method for monitoring the conjugation reaction of peptides to the carrier mannan; (KG)_5_-MOG_35-55_ was used as the peptide example in this study. Peptide-OM conjugates can serve as potential vaccine candidates as has previously been shown by the group for cancer models and more recently in MS models. Electrochemical voltammetric techniques and HPLC experiments were used to confirm the conjugation of (KG)_5_-MOG_35-55_ to the aldehyde groups of OM. It is shown that voltammetric technique and HPLC can be used to monitor the conjugation efficiency of peptide-carrier conjugates.

## Figures and Tables

**Figure 1 brainsci-10-00577-f001:**
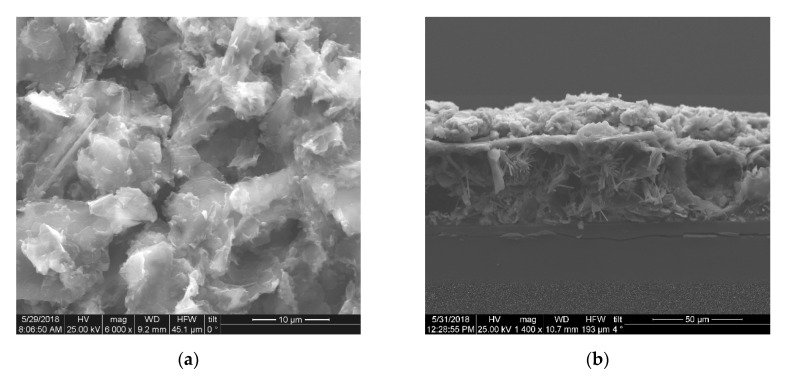
SEM images of the graphite/SiO_2_ working electrode from (**a**) top view and (**b**) a cross section.

**Figure 2 brainsci-10-00577-f002:**
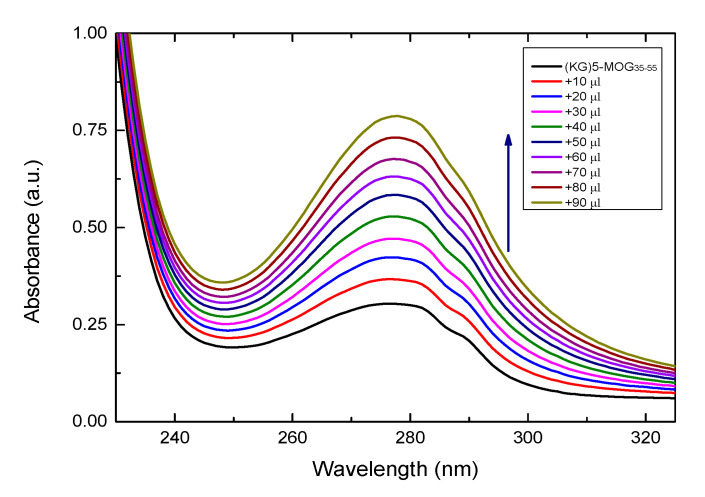
UV-Vis spectral changes of (KG)_5_-MOG_35-55_ in solution with increasing amounts of OM (10–90 μL).

**Figure 3 brainsci-10-00577-f003:**
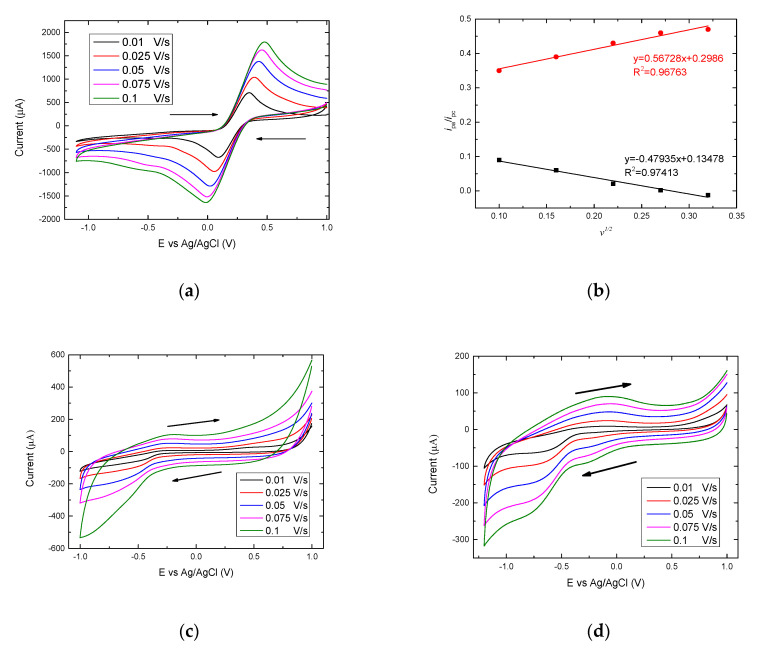
Cyclic voltammograms (CVs) of (**a**) a bare graphite/SiO_2_ film electrode in 0.1 M KCl solution containing 5 mM of [Fe(CN)6]^3−/4−^ at different scan rates. (**b**) Plot of anodic and cathodic peak current (Ipa/Ipc) vs. square root of scan rate (*v*^1/2^). (**c**) A bare graphite/SiO_2_ film electrode in 10 mM NaH_2_PO_4_, pH 7.0 at different scan rates and (**d**) the OM-(KG)_5_-MOG_35-55_ conjugate on graphite/SiO_2_ in 10 mM NaH_2_PO_4_, pH 7.0 at different scan rates under an Argon atmosphere.

**Figure 4 brainsci-10-00577-f004:**
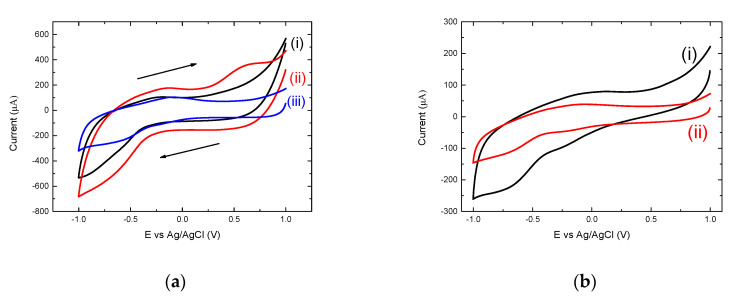
(**a**) CV scans at a scan rate of 0.1 Vs^−1^ of (i) a bare graphite/SiO_2_ film electrode, (ii) mannan and (iii) OM-(KG)_5_-MOG_35-55_ conjugate. (**b**) Depicts the comparison between the CV’s of (i) MOG_35-55_ and (ii) MOG_37-55_, both on graphite/SiO_2_ in 10 mM NaH_2_PO4, pH 7.0 at scan rate of 0.075 Vs^−1^.

**Figure 5 brainsci-10-00577-f005:**
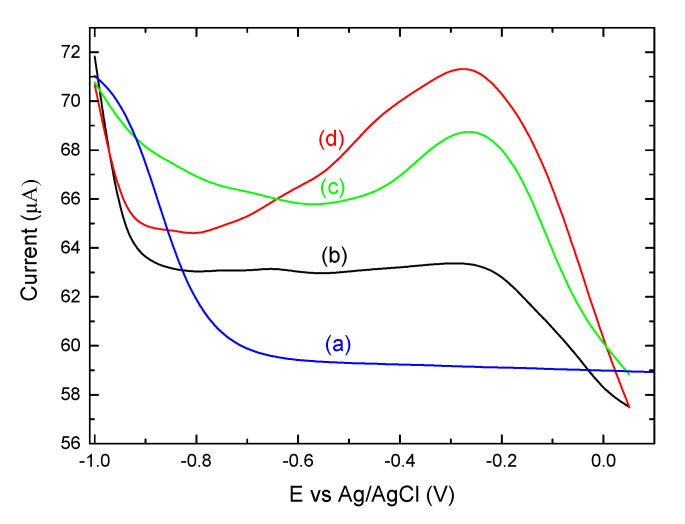
Differential pulse voltammetry (DPVs) comparison of (a) a bare graphite/SiO_2_ film electrode, (b) (KG)_5_-MOG_35-55_, (c) mannan, and (d) OM-(KG)_5_-MOG_35-55_ conjugate on graphite/SiO_2_ electrode in 10 mM NaH_2_PO_4_, pH 7.0.

**Figure 6 brainsci-10-00577-f006:**
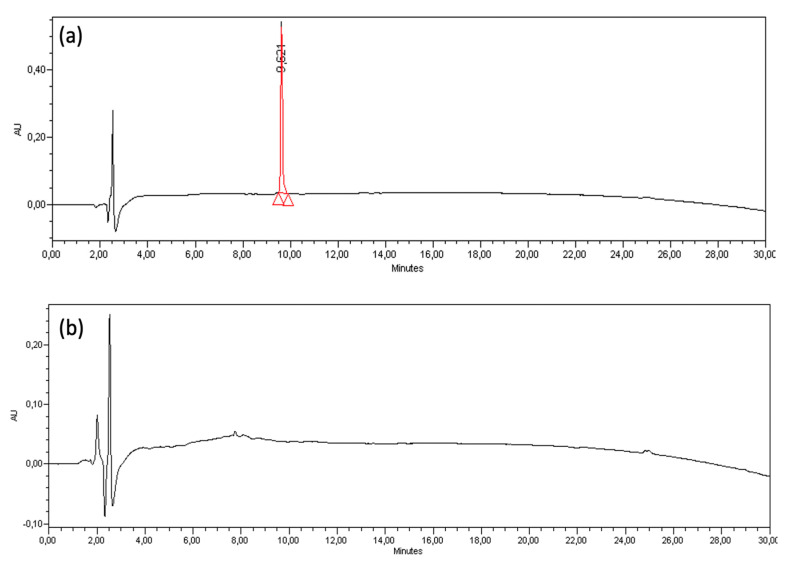
(**a**) High-performance liquid chromatography (HPLC) analysis of (KG)_5_-MOG_35-55_–214 nm at the beginning of the conjugation reaction and (**b**) HPLC analysis of OM-(KG)_5_-MOG_35-55_ solution after 6 h.

**Figure 7 brainsci-10-00577-f007:**
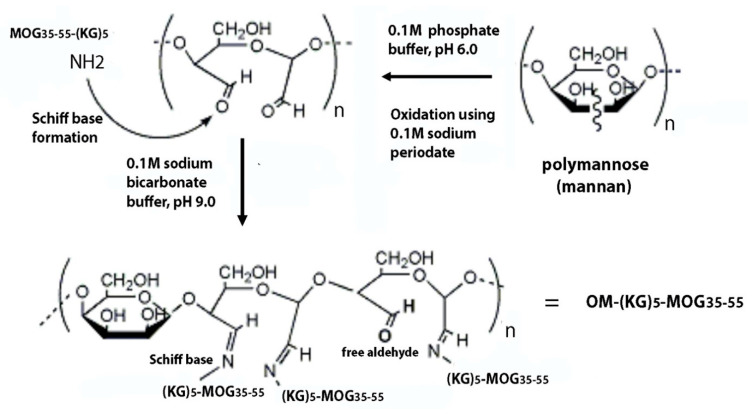
The mechanism of cis diol cleavage. Synthetic scheme of conjugation reaction of peptide with oxidized mannan [35].

**Table 1 brainsci-10-00577-t001:** Peptides and conjugates used in this study.

Acronym	Specification
MOG_35-55_	Myelin oligodendrocyte glycoprotein immunogenic epitope, region 35–55
MOG_37-55_	Myelin oligodendrocyte glycoprotein immunogenic epitope, region 37–55
(KG)_5_-MOG_35-55_	Peptide analogue MOG_35-55_ with (KG)_5_ at the N-terminus
OM-(KG)_5_-MOG_35-55_	Oxidized mannan conjugated to (KG)_5_-MOG_35-55_

KG, lysine glycine; MOG, myelin oligodendrocyte glycoprotein; OM, oxidized mannan.

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
