# Peer review of "The Use of Electrochemical Voltammetric Techniques and High-Pressure Liquid Chromatography to Evaluate Conjugation Efficiency of Multiple Sclerosis Peptide-Carrier Conjugates"

_brainsci, 2020, doi:10.3390/brainsci10090577_

Round 1

Reviewer 1 Report

Novel techniques for drug detection to examine conjugation of peptides relevant to treat MS. Some editing of the manuscript is required

Line 23 Change to: In our recent studies,

26 change to detected as a conjugate

39 change to: dependent on

44 change to: exhibit a wide

64 change to: to the mannose

68 change to: PLP 139-141

93/94 change to: Saccharomyces cerevisiae

127 change to: to over 97% purity

134 change to: Saccharomyces cerevisiae

139 change to: 10.5 rather than 10,5 (and correct throughout text)

273 change to: In contrast to (this sentence needs re writing as it is not clear)

290 change to: 35 (remove x)

300 change to: to block further

Reviewer 2 Report

In the present study Deskoulidis E at al., have described the conjugation of the multiple sclerosis epitope peptide MOG35-55 with mannan using electrochemical voltammetric and HPLC techniques, an alternative to lengthy capillary electrophoresis and polyacrylamide gel electrophoresis.

Major Comments:

  1. HPLC technique alone itself can confirm the conjugation of MOG35-55 with mannan. In such case reviewer is not convinced that there is an  unmet need to develop methods like electrochemical voltammetric to monitor conjugation of sclerosis epitope peptide MOG35-55 with mannan.
  2. Can we quantitate the oxidized mannan-[Lys-Gly]5-MOG35-55 conjugate using electrochemical voltammetric technique? MS, HPLC or fluorescent based techniques can confirm the conjugation quantitatively. 
  3. What is the sensitivity of DPV technique to detect the least amount of API, in this case oxidized mannan-[Lys-Gly]5-MOG35-55 conjugate?
  4. Overall reviewer is not convinced about the importance of electrochemical voltammetric technique over capillary electrophoresis or poly acrylamide gel electrophoresis. 

Minor Comments

  1. Typo- Line 167- Characterization
  2. All the different MOG peptide sequences to be tabulated.

Round 2

Reviewer 2 Report

Thank you for responding to all the comments satisfactorily.

Authors contribution towards developing this voltametric technique as an alternative to the available HPLC/MS/PAGE techniques to monitor the mannan-peptide conjugates could possibly lead the API industry to adopt this as a standard technique which can in future replace the existing methods.